TECHNICAL RELEASE

# Optimizing experimental design for genome sequencing and assembly with Oxford Nanopore Technologies

John M. Sutton[1], Joshua D. Millwood[1], A. Case McCormack[1] and Janna L. Fierst[1,*]

1 Department of Biological Sciences, University of Alabama, Tuscaloosa, AL 35487-0344, USA

## ABSTRACT

High quality reference genome sequences are the core of modern genomics. Oxford Nanopore Technologies (ONT) produces inexpensive DNA sequences, but has high error rates, which make sequence assembly and analysis difficult as genome size and complexity increases. Robust experimental design is necessary for ONT genome sequencing and assembly, but few studies have addressed eukaryotic organisms. Here, we present novel results using simulated and empirical ONT and DNA libraries to identify best practices for sequencing and assembly for several model species. We find that the unique error structure of ONT libraries causes errors to accumulate and assembly statistics plateau as sequence depth increases. High-quality assembled eukaryotic sequences require high-molecular-weight DNA extractions that increase sequence read length, and computational protocols that reduce error through pre-assembly correction and read selection. Our quantitative results will be helpful for researchers seeking guidance for *de novo* assembly projects.

**Subjects**  Animal and Plant Sciences, Molecular Genetics, Bioinformatics

# DATA DESCRIPTION

## Background

Many factors affect the quality of a *de novo* genome assembly. Genome size increases the size of the "puzzle" to put together, while the size of the pieces (sequence reads) remains the same. Repetitive regions and mobile genetic elements create unique challenges because they are present in more than one location in the genome, so without contextual information it is difficult to identify how many copies exist in the complete genome. For example, *Alu* repeat elements reach >1 million copies in the human genome [1].

Third generation sequencing technologies can theoretically solve these problems. Long sequence reads span repetitive regions and can potentially identify the exact size and location of repeats on a chromosome. The larger "puzzle pieces" require less sequencing effort to span the entire genome. Pacific Biosciences (PacBio) and Oxford Nanopore Technologies (ONT) are the current front-runners in long-read sequencing platforms; both are capable of average read lengths in the order of tens of thousands of base pairs and, theoretically, entire chromosomes can be sequenced in a single read [2, 3]. Both are also capable of producing high-quality assembled sequences with reasonable amounts of data [4].

**Submitted:**   17 February 2021

\* Corresponding author. E-mail: janna.l.fierst@ua.edu

Preprint submitted at https://doi.org/10.1101/2020.05.05.079327

ONT offers several advantages over PacBio. Nanopore sequencing relies on running molecular fragments through engineered nanopores and recording the resulting alterations in electrical current. The technology is versatile and can be used for DNA sequencing, mRNA sequencing, amplification-free mRNA quantification [5], and measuring DNA base modifications like methylation [6, 7]. ONT libraries can be readily prepared with low amounts of input DNA, which is an important consideration when studying small or difficult-to-sample organisms. ONT platforms are inexpensive and designed to be used in individual research laboratories and classrooms [8]. Both PacBio and ONT continually refine their technologies and, we believe, both will be a part of modern science. Similar projects have analyzed experimental design for PacBio [9, 10] but the few benchmarking studies for ONT sequencing and assembly have been carried out with prokaryotes [11, 12] or single organisms with large, repeat-heavy genomes [13]. For these reasons, we have chosen to study ONT and quantify how this inexpensive, accessible technology may be best utilized to produce high-quality assembled reference genome sequences. Here, we fill an important gap in the literature by addressing eukaryotic organisms with moderately sized genomes.

ONT sequence reads present unique challenges for genome sequence assembly. DNA molecules do not move through protein nanopores at a constant rate, and changes in current are a composite signature reflecting 3–5 nucleotides occupying the nanopore (for R9.4.1 flow cells). The signal processing has trouble detecting changes in current with homopolymers (single nucleotide repeats, for example AAAAAA), short tandem repeats and heavily methylated sites [14, 15]. As a result, ONT sequence reads contain small nucleotide stretches that have been incorrectly identified, inserted and deleted (Figure 1A, B). This error structure results in relatively few large, contiguous stretches of correctly identified nucleotides (Figure 1B) for prokaryotes and eukaryotes (Figure 1C, D) and is uniquely challenging for assembly algorithms.

## Context

The typical recommendation for genome assembly is that increasing sequencing depth (i.e., the average number of times a nucleotide in the genome is sequenced) increases assembly quality and contiguity. Here, we test this by simulating extremely high depth ONT libraries for *Escherichia coli* (NCBI:txid562), *Caenorhabditis elegans* (NCBI:txid6239) roundworms, *Arabidopsis thaliana* (NCBI:txid3702) mouse-ear cress, and *Drosophila melanogaster* (NCBI:txid7227) fruit flies. For each organism, we assemble sequence read sets at different depths and measure the contiguity, completeness, and accuracy of assembled sequences relative to the current reference genome. Many *de novo* assembly projects target organisms without reference genomes, and we also measure the identification of a set of genes thought to be found in single copy in each organism. Pure ONT data sets result in superior assembled genome sequences, but can be expensive to generate. We also analyze assembled sequences from "hybrid" DNA read data sets comprising ONT and less expensive short Illumina sequence reads.

We use results from our simulated sequences to guide experimental ONT sequencing and assembly for four strains of the nematodes *C. remanei* and *C. latens*. For eukaryotes, both ploidy and breeding system can influence the assembly of genome sequence. Polyploidy can create scenarios in which many sites in a genome look similar, making it difficult to place these regions within an assembled genome [16, 17]. For non-haploid organisms, there

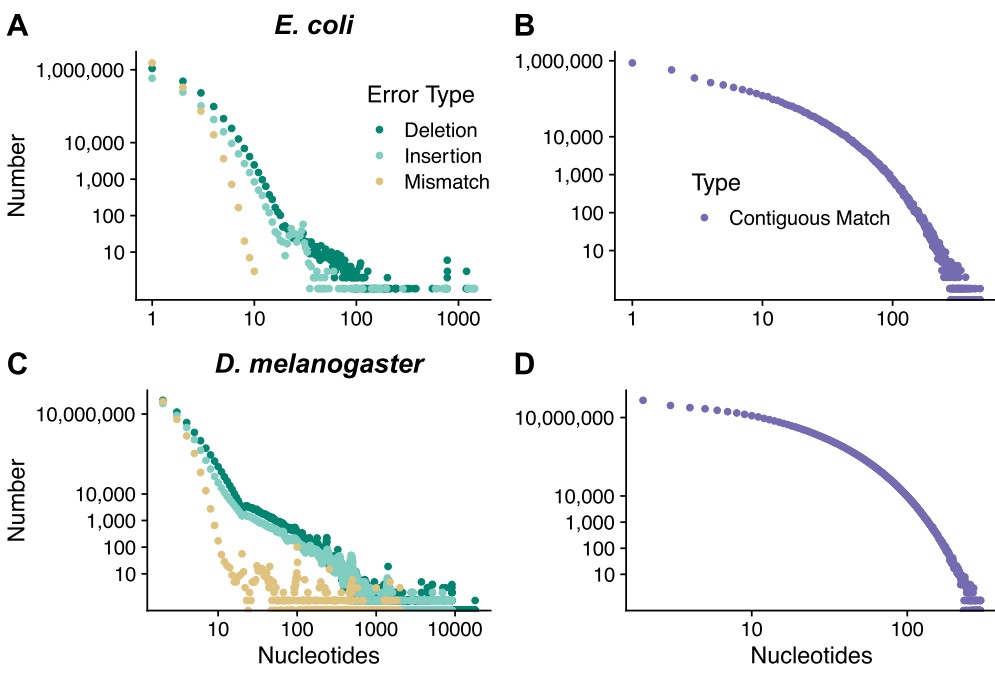

**Figure 1.** ONT sequence reads contain mixtures of errors including miscalled nucleotides, deletions, insertions and truncated homopolymers. When aligned to the reference genome this results in many single and multi-base deletions, insertions and mismatches for (A) *E. coli* and (B) relatively few stretches of contiguous matching sequence that extend beyond a few nucleotides. For (C), *D. melanogaster*, the error profile is similar but the large, repeat-rich genome results in multi-nucleotide deletions and insertions, and again, in (D), few stretches of long, contiguous matching sequence.

may be heterozygosity, and at these sites the effective sequencing coverage is cut in half. For small, non-clonal metazoans, sufficient DNA for sequencing cannot be acquired from a single individual. DNA must be extracted from pools of individuals, and – for outbreeding organisms – there may be substantial individual diversity within the pool.

We find that ONT approaches can produce contiguous assembled sequences with relatively high sequencing coverage of >60× – well above current recommendations [18]. Pure ONT sequencing and assembly outperforms the hybrid approach. We find that contiguous assemblies cannot be achieved by solely increasing ONT sequencing depth as errors accumulate and assembly statistics plateau. Pre-assembly filtering and read correction improve contiguity, and post-assembly polishing using short Illumina DNA sequence reads increases accuracy. We also find that the use of Illumina data, even at low sequencing depths, increases accuracy through iterative polishing. Our results demonstrate that rigorous experimental design can improve inference of a genome sequence and reduce the effort and cost required.

## METHODS

### DNA extraction and sequencing

Nematodes used for genomic sequencing were grown on two 100-mm nematode growth medium (NGM) plates [19] seeded with *E. coli* OP50. Worms were harvested by washing plates with M9 mineral medium into 15 mL conical tubes. Samples were rocked on a tabletop rocker for 1 hour before being centrifuged to pellet worms. The supernatant was

removed and tubes refilled with sterile M9 mineral medium, mixed and pelleted by centrifugation again. This process was repeated five times or until the supernatant was clear after centrifugation. The pellet was moved to 2-mL tubes and frozen at −20 ˚C until extraction. Worm pellets were allowed to thaw to room temperature, then flash-frozen in liquid nitrogen and repeated three times. Worm pellets were placed in 1.2 mL of lysis buffer solution (100 mM ethylenediaminetetraacetic acid [EDTA], 50 mM Tris, and 1% SDS) and 20 μL of Proteinase K (100 mg/mL). Tubes were then placed on a 56 ˚C heat block for 30 minutes with shaking. Genomic DNA (gDNA) was then isolated using a modified phenol-chloroform extraction [20]. DNA concentrations and purity were measured with a Qubit 4 (Life Technologies, Carlsbad, CA USA) and NanoDrop® 1000 spectrophotometer, respectively (Thermo Fisher Scientific, Waltham, MA USA). Extracts were visualized on a 0.8% agarose gel to verify high-molecular-weight gDNA. DNA size selection was carried out using the Short Read Eliminator Kit from Circulomics Inc., according to the manufacturer's guidelines (Baltimore, MD, USA).

DNA libraries were prepared for each sample using the SQK-LSK109 ligation sequencing kit and loaded on to R9.4.1 RevD flow cells. The recommended protocol from ONT was modified by replacing the first AmpureXP bead clean step with an additional treatment with the Short Read Eliminator Kit. Approximately 700 ng of gDNA from each library was loaded on to a flow cell sequenced for 48 hours on a GridION X5 platform with basecalling performed by Guppy v.4.0.11 (Oxford Nanopore Technologies) set to high-accuracy mode. ONT reads for each strain were trimmed of adapters using Porechop (RRID:SCR_016967) [21] with the – discard_middle flag to remove chimeric reads.

## Assembly and decontamination of real data sets

We evaluated contaminants in the *C. remanei* PX356, *C. remanei* PX439, *C. remanei* EM464 and *C. latens* PX534 draft assemblies with Blobtools2 [22] and SIDR [23]. Blobtools2 uses phylum-level taxonomic assignment, read coverage depth and GC content to identify non-target contigs within the assembly. To prepare data for visualization with BlobToolKit, the ONT reads were aligned to the initial assemblies using minimap2 v2.17 [24] and sorted using SAMtools v1.8 (RRID:SCR_002105) [25]. A reference database for taxonomic identification of scaffolds was created using blastn 2.2.31 (RRID:SCR_001598) [26] using the Nucleotide database from the National Center for Biotechnology Information [27]. An additional database was generated using diamond tblastx (RRID:SCR_011823) [28]. Contigs that were >10,000 bp and taxonomically identified as *Nematoda* were retained in the assembly.

SIDR [23] utilizes ensemble-based machine learning to train a model capable of discriminating target and contaminant contigs based on measured predictor variables. To prepare data for SIDR decontamination Illumina, DNA and RNA sequence reads were aligned to the draft assembly with bwa [29] using the BWA-MEM algorithm. We used bbtools/bbmap software (RRID:SCR_016965) [30] to calculate the average sequence coverage, length, GC, bases covered in alignment, RNA average sequence coverage and bases covered in RNA alignment for each contig. We used these data to train a bootstrap aggregated (bagged) decision tree model based on the blast identification of contig origin and used this model to predict the origin, target or contaminant of each contig. SIDR allowed us to assign probable *Nematoda* origin to contigs that lacked blast identification. We discarded contaminants common to *Nematoda* genome assemblies including *E. coli*,



*Pseudomonas*, *Serratia* and *Stenotrophomonas* [31]. We also identified one large 4.57 Mb *E. coli* contig in the *C. latens* PX534 that had been labeled *Nematoda* by BlobToolKit and removed it from the *C. latens* PX534 assembled sequences.

## Heterozygosity analyses

We assessed heterozygosity using the software Jellyfish v2.2.4 (RRID:SCR_005491) [32] and GenomeScope (RRID:SCR_017014) [33]. Jellyfish is a fast, memory efficient *k*-mer counting tool that searches for occurrences of unique *k*-mers in raw sequence reads based on user selected size. GenomeScope estimates genome heterozygosity, genome size, and repeat content using a *k*-mer-based statistical approach. Jellyfish was run using the default settings and a *k*-mer size of 21. This *k*-mer size was selected based on recommendation from the maker of Jellyfish and should be sufficient to obtain all unique *k*-mers present. A histogram file of all *k*-mer occurrences was created by Jellyfish and then uploaded into the online interface of GenomeScope. GenomeScope was run using default settings with a *k*-mer length of 21 and a read length consistent with each sample's sequencing parameters.

## Evaluation

We used the software BUSCO version 4.0.1 (Benchmarking Universal Single-Copy Orthologs; RRID:SCR_015008) [34] to identify conserved gene sets. Briefly, BUSCO searches assembled DNA sequences for a set of unique genes that are expected to be conserved in single copy in an evolutionarily related group of organisms. We used BUSCO v4.0.1 [34] and the data sets Nematoda_odb10 for *C. elegans*, *C. remanei*, and *C. latens*; Metazoan_odb10 for *D. melanogaster*; and Viridiplantae_odb10 for *A. thaliana*. We also measured BUSCO completeness with the Diptera_odb10 for the *D. melanogaster* assemblies, but found that, in some instances, our assembled sequence contained a greater proportion of conserved genes than the reference sequence. We chose to focus on the Metazoan_odb10 for *D. melanogaster* and present both sets of statistics in the additional files in the *Gigascience* database [35]. BUSCO genes in each assembly were classified as single copy, duplicated, fragmented or missing. For single-copy and duplicated classifications, the target gene must be present and >95% identical to the expected size and sequence of the reference in the database [34].

## Simulated sequence libraries

We obtained ONT sequences (R9 chemistry) from the National Center for Biotechnology Information (NCBI) Sequence Read Archive (SRA) [36]. The *E. coli* data set contained 11,652,330 sequenced bases in 120,151 reads (low ~2.5× coverage of the 4.64-megabase pair [Mb] reference sequence), the *C. elegans* data set contained 8,860,671,330 sequenced bases in 583,466 reads (~87.9× coverage of the 100.8-Mb reference sequence), the *A. thaliana* data set contained 3,421,779,258 sequenced bases in 300,071 reads (~25.22× coverage of the 135.67-Mb reference sequence), and the *D. melanogaster* data set contained 4,617,842,308 sequenced bases in 663,784 reads (~32.39× coverage of the 142.57-Mb reference sequence). We obtained the reference genome sequence for *E. coli* strain K12_MG1655 from NCBI; all other reference genome sequences were obtained from Ensembl (release 95) [37].

We simulated 150 base pair (bp) paired-end Illumina DNA libraries with the software ART (RRID:SCR_006538) [38] and ONT DNA libraries with the software NanoSim v2.0.0 (RRID:SCR_018243) [39]. Both software programs utilize an assembled sequence to generate simulated libraries with read profiles like an empirically generated library. NanoSim



**Table 1.** ONT flow cells, reads and data produced.

| Organism | Flow Cell ID | Total #Reads (K) | Total bp passed (Gb) |
|---|---|---|---|
| *C. remanei* PX464 | 464.1 | 391 | 4.11 |
| | 464.2 | 457 | 5.39 |
| *C. latens* PX534 | 534.1 | 545 | 0.579 |
| | 534.2 | 504 | 0.784 |
| | 534.3 | 492 | 2.76 |
| | 534.4 | 824 | 2.35 |
| | 534.5 | 719 | 4.35 |
| | 534.6 | 1020 | 5.41 |
| *C. remanei* PX439 | 439.1 | 2080 | 6.1 |
| | 439.2 | 940 | 5.3 |
| | 439.3 | 398 | 5.59 |
| *C. remanei* PX356 | 356.1* | 2240 | 4.31 |
| | 356.2 | 940 | 5.01 |
| | 356.3 | 1048 | 4.19 |
| | 356.4 | 750 | 0.723 |
| | 356.5 | 1710 | 4.16 |

*Sequenced with ONT SQK-LSK108 Library Prep Kit.

requires real ONT flowcell data to simulate organism-specific mismatch, insertion and deletion rates. ART libraries were simulated to a coverage of 300× with a fragment standard deviation of 50 bp. NanoSim libraries mirror the characteristics of the real ONT flowcells including read length distributions. We used NanoSim to simulate 500,000 *E. coli* reads (943× coverage; read N50 8158), 2,000,000 *C. elegans* reads (336× coverage; N50 21,559), 5,000,000 *A. thaliana* reads (420× coverage; N50 19,577) and 10,000,000 *D. melanogaster* reads (219× coverage; N50 11,955). The sequence read N50 indicates that 50% of the total sequenced nucleotides are in reads of that length or longer. Libraries and descriptive statistics including mean read lengths and qualities are available in the *Gigascience* database [35]. Using simulated libraries allowed us to study depth beyond the capability of single ONT flowcells, and to avoid possible errors created by combining libraries generated by different labs under different conditions. A similar approach utilizing simulated libraries was used by Wick and Holt [12] to study prokaryotic genome sequences.

## *C. remanei* and *C. latens* sequence libraries

Output for each flow cell used for each strain is summarized in Table 1. Based on an estimated genome size of 125 Mb [40], we generated 76–156× coverage of each strain with sequence read N50 8203–22,891 bp.

## Genome assembly
### Assembly of simulated data

We analyzed ONT libraries assembled with the Canu (RRID:SCR_015880) [18] and Flye (RRID:SCR_017016) [41] software packages. Both are well-established assemblers that consistently outperform other approaches in assembly contiguity and accuracy [11–13]. We used a 64-core 512 gigabyte (GB) Dell PowerEdge R730 server for all assemblies and found these resources sufficient for our target organisms and libraries. Other assembly software, such as Miniasm/Minipolish [42], wtdbg2 (RRID:SCR_017225)/redbean [43], Shasta [44] and Raven (RRID:SCR_001937) [45], prioritize computational efficiency and may be appropriate for large genomes or laboratories with computational limitations.

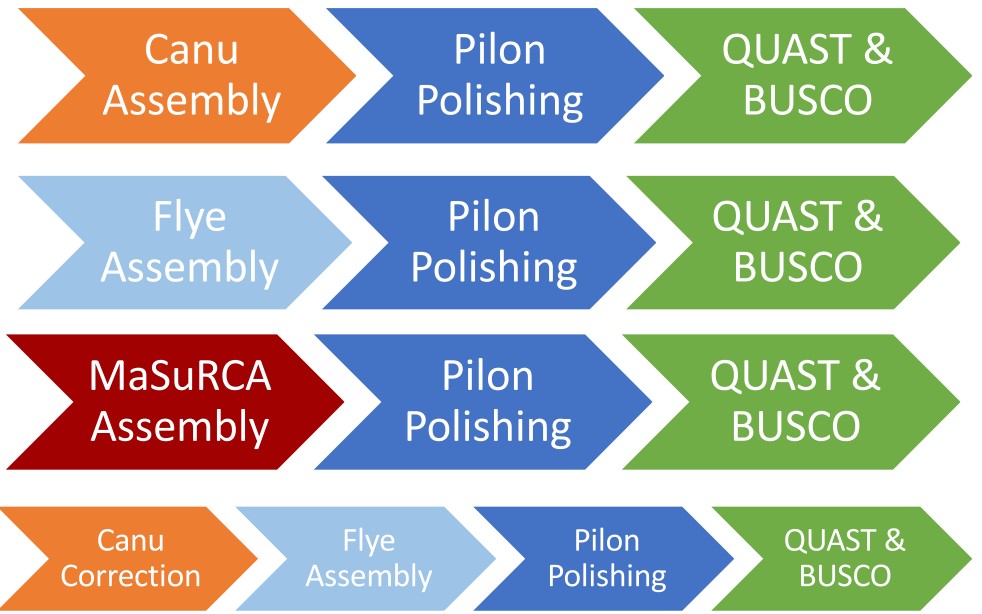

**Figure 2.** Assembly, polishing and measurement approaches used in this study.

We assembled genome sequences using three approaches (Figure 2). We first assembled the simulated ONT read sets with Canu v1.9 [18]. Each genome was assembled at decreasing coverage depths until the assembler errored out (i.e., failed to produce an assembly) or produced an assembly far from the appropriate size for the organism. To minimize the influence of individual reads and stochastic assembly artifacts, each read set was generated by selecting a random subset of the full simulated data set. The second approach used the same subsets of ONT data but was assembled with Flye 2.8 [41].

Hybrid genome assembly combining high-accuracy Illumina short reads with ONT long reads can be performed with Unicycler [46] or MaSuRCA (Maryland Super Read Cabog Assembler; RRID:SCR_010691) [47]. Unicycler is currently only recommended for bacterial genomes, so we focus here on MaSuRCA performance. The long-read data sets that produced the most contiguous, highest accuracy genome sequences were paired with 2 ×150 simulated paired-end Illumina data and assembled using MaSuRCA v3.3.9 [48]. Coverage depths were adjusted for both data sets to better understand the effects of increasing or decreasing coverage on the final assembly. Here, we retained the ONT data set to maximize our ability to draw parallels between assembly approaches. For example, the minimum *C. elegans* ONT data set that assembled with Canu [18] was an average of ~60× coverage across the genome. This read set was used in the MaSuRCA trials, with 50× and 100× Illumina coverage, respectively.

We also used Flye [41] to assemble read sets that had been corrected using the "canu – correct" function. Canu correction uses all-versus-all overlap information to correct individual reads [18]. In addition to the consensus information, Canu employs two filters to avoid biases caused by sequence quality or repeats. From read-length estimates, Canu uses the longest available reads up to the user-specified coverage depth for correction [18]. By default, this is set to 40× coverage. Assembling these data sets with both Canu and Flye



allowed us to compare the effectiveness of the error correction modules within each software package. We also assembled libraries with reads selected by the Canu correction module but not subject to correction to test the influence of error correction versus read length on assembly.

The most contiguous assemblies from the long-read only and hybrid categories were error corrected using Pilon v1.23 (RRID:SCR_014731) [49] to determine the effect of short-read polishing on the accuracy of the draft assemblies. Each simulated assembled sequence was polished with the entire simulated paired-end data set. Four rounds of polishing were completed for each assembly, with statistics measured after each round with QUAST v5.0.2 (RRID:SCR_001228) [50] and BUSCO v4.0.3 [34].

### Assembly of ONT nematode data

Preliminary assemblies were created for each strain by first correcting the complete library with Canu – correct (Canu v2.0) [18]. Corrected reads from Canu were then assembled using Flye 2.8 [40]. The draft assembly for each strain was polished using Pilon v1.23 [49] and Illumina paired-end reads. All data sets contained a small subset of nematode microbiota that were removed from the final assemblies [22, 23].

## DATA VALIDATION AND QUALITY CONTROL
## Assessing the quality of assembled simulated sequences

We measured genome statistics relative to the reference sequence of each organism with QUAST [50]. We assessed contiguity and accuracy of the assembled sequence through eight statistics:

1. NG50 is a size median statistic indicating that 50% of the expected assembled genome sequence (where the expectation is based on a known reference) is contained in contiguous sequences of that size or larger.
2. NGA50 is a similar size median, but indicates that 50% of the expected assembled genome sequence that aligns to the reference genome is contained in contiguous sequences of that size or larger.
3. LG50 is the number of linkage groups or contiguous assembled sequences containing 50% of the expected assembled genome sequence.
4. LGA50 is similar but is measured in the portion of the assembled sequence that aligns to the reference genome.
5. Genome fraction (%) is the fraction of reference genome captured in the assembled sequence.
6. Duplication measures the fraction of reference genome found multiple times in the assembled sequence.
7. Mismatches is the number of mismatched nucleotides per 100,000 nucleotides or base pairs (bp).
8. Indels are the number of insertions and/or deletions per 100,000 nucleotides.

For our metazoan organisms we also used the software package BUSCO v4.0.3 [34] to search for a set of unique genes expected to be conserved in single copy in an evolutionarily related group of organisms.



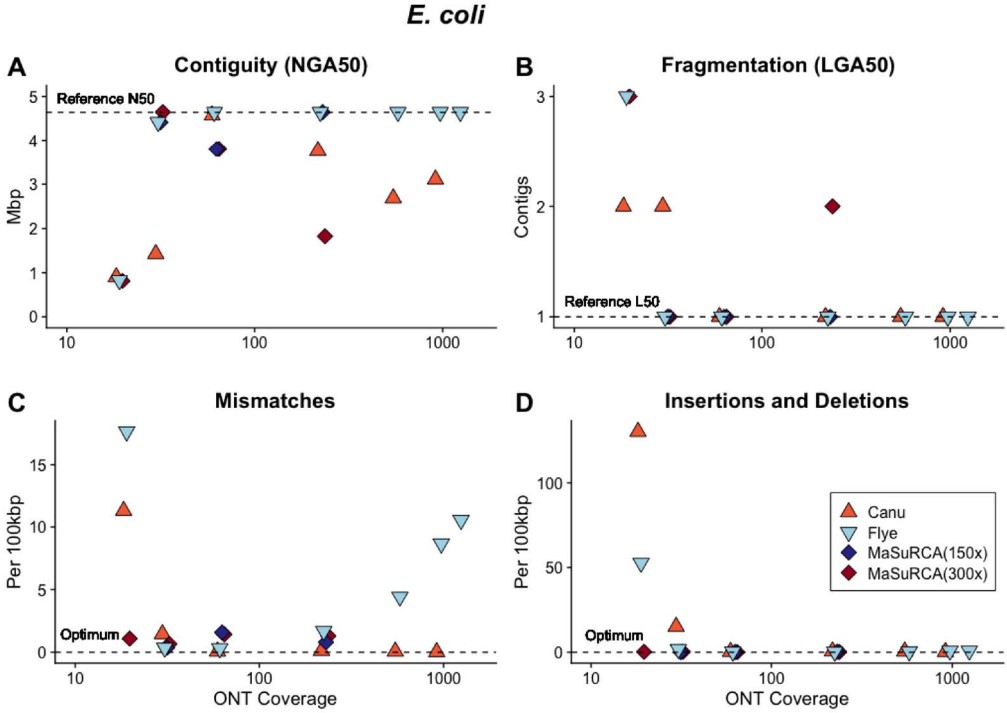

**Figure 3.** The *E. coli* genome is relatively small at 4.64 Mb, and is less complex than metazoan genome sequences. Assembly of ONT libraries at relatively high coverage (>60× average sequence depth) with Canu and Flye results in assembled sequences with (A) high contiguity; (B) low contig number converging on a single chromosome; (C) few mismatches; and (D) few insertion/deletion errors (indels) than the reference sequence. The influence of sequencing coverage on contiguity or fragmentation was not monotonic for Canu- or MaSuRCA-assembled sequences. The relationship between sequencing coverage and accuracy was not monotonic for Flye assembled sequences.

## *Escherichia coli*

The *E. coli* genome is relatively small at 4.64 Mb and contains few repeats (87.8% of the genome codes for proteins [51]). Assembly of 50,000 ONT sequences (~62× coverage; N50 8350 bp) with Canu [18] and Flye [41] resulted in circular contigs with high contiguity and few mismatches, insertions or deletions (Figure 3A–D). At low ONT coverage (~19×), the assemblies produced by Canu and Flye were contained in six and nine contigs, respectively. Both assembled sequences had high rates of mismatches, insertions and deletions than sequences assembled from high coverage libraries (Figure 3C, 3D). Increasing ONT coverage beyond 62× resulted in poor contiguity for Canu-assembled sequences (Figure 3A) and increased mismatches for Flye-assembled sequences (Figure 3C). Assembly of hybrid Illumina–ONT libraries with MaSuRCA [48] resulted in sequences with variable contiguity and accuracy. Two of the tested hybrid sets could assemble the genome into a single contig, but the library with the highest Illumina coverage (300×) and highest ONT coverage (227×) assembled into two contigs. MaSuRCA was unable to perform as well as either Canu or Flye when given the same long-read data set. For instance, the top performing Canu and Flye runs used 50,000 ONT reads (~62× coverage); the same reads passed through MaSuRCA produced two contigs, regardless of Illumina coverage (Figure 3B).

After polishing the assemblies with Pilon [49], the Canu-assembled sequence had fewer mismatches and indels per 100 kilobase pairs (Kb) than the Flye-assembled sequence or an

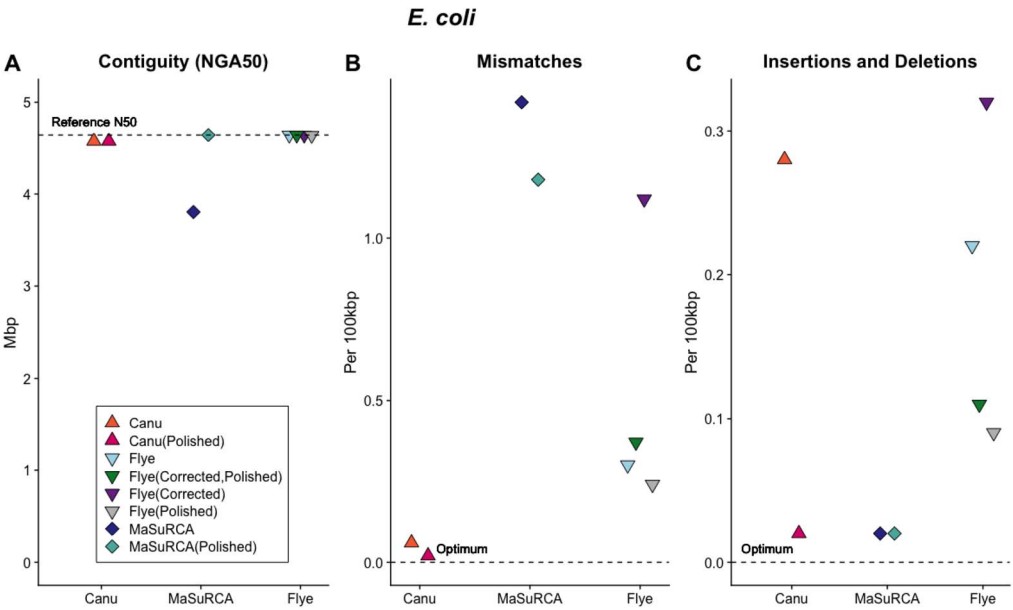

**Figure 4.** The relative performance of Canu, Flye and MaSuRCA assembly software with raw data sets, data sets with reads pre-corrected ("Corrected" in the figure legend) and post-assembly polishing with Illumina short reads ("Polished" in the figure legend) as measured by (A) genome contiguity; (B) mismatches and (C) insertions and deletions relative to the *E. coli* reference sequence. Both Canu and Flye assembled highly contiguous sequences for all data sets and the MaSuRCA assembled sequence was highly contiguous post-polishing. The Canu assembled sequences had few mismatches and insertion/deletion errors were eliminated through polishing. The MaSuRCA assembled sequence had numerous mismatches that were not addressed through polishing and few insertions and deletions. The accuracy of all Flye assembled sequences was improved with polishing.

assembly produced by Flye using reads that were first corrected with Canu (Figure 4A–C). Despite its dependence on higher accuracy Illumina sequences, the MaSuRCA assembled sequence had more mismatches, insertions and deletions both before and after polishing (Figure 4B). We used MaSuRCA to assemble a set of paired-end Illumina sequences to test the influence of adding ONT data to the assembly process. The resulting sequence was 98.76% of the *E. coli* genome and was fragmented into 74 contiguous pieces. The accuracy, 1.7 mismatches and 0.04 indels per 100 Kb, was improved versus the hybrid assembled sequences. This indicates that the inclusion of ONT data can introduce errors in hybrid approaches that cannot be corrected with post-assembly polishing.

## *Caenorhabditis elegans*

The *C. elegans* genome is 100.8 Mb [52] contained in a single X chromosome and five autosomes [53]. *C. elegans* is a diploid self-fertile hermaphrodite with low levels of genetic diversity [54]. Approximately 16% of the genome is repetitive [55]. Canu, Flye, and MaSuRCA-assembled sequences plateaued in contiguity and accuracy above ~70× coverage (Figure 5A–D). The read N50 for these data sets ranged between 20,742 and 22,840. Both Canu and Flye assembled chromosome-scale contigs with low rates of mismatches, insertions and deletions. The MaSuRCA [48] hybrid assembly approach did not perform as well, even with high ONT coverage (Figure 5A, B).

To improve these assemblies, we experimented with polishing the Canu and Flye assembled sequences. Additionally, we used Flye to assemble the Canu-corrected data set

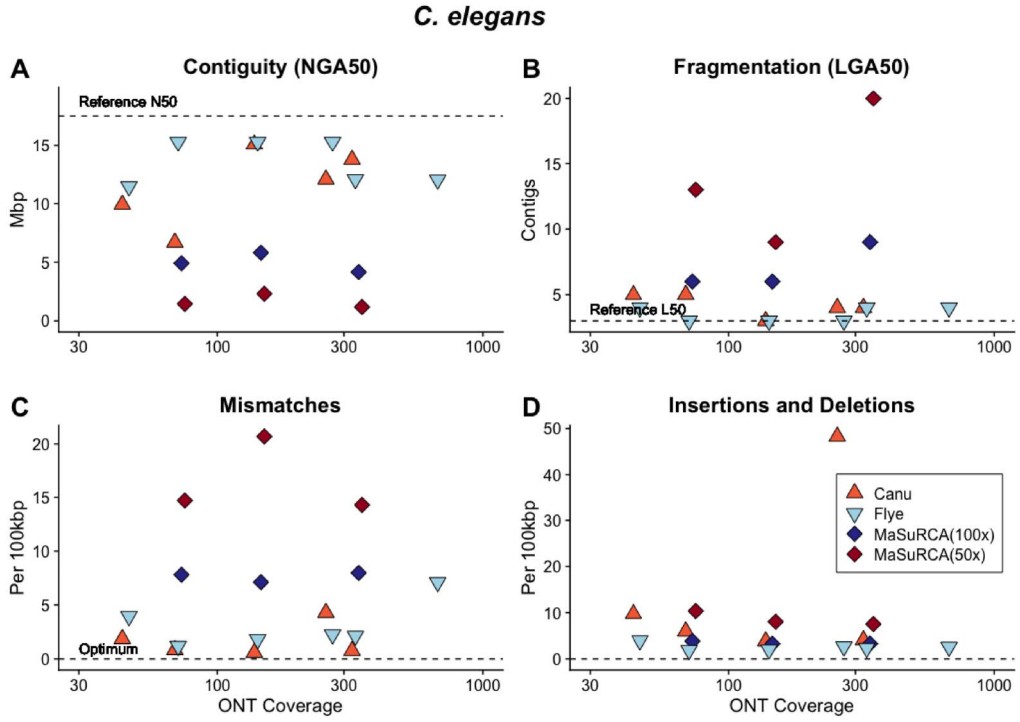

**Figure 5.** Both Canu- and Flye-assembled libraries with >70× ONT coverage into genome sequences with (A) high contiguity; (B) low fragmentation; (C) few mismatches; and (D) few insertions and deletions compared with the *C. elegans* reference sequence. In comparison, MaSuRCA performed poorly across a range of ONT sequencing coverages.

("Corrected" in Figure 6) and used Flye to assemble the reads selected by the Canu correction module but not corrected ("Selected" in Figure 6). The data set corrected with Canu and assembled with Flye had a read N50 of 28,641 bp. It produced six chromosome-scale contigs with low numbers of mismatches, insertions and deletions, which further decreased after polishing (Figure 6A–D). Flye assembly of the same data set prior to correction resulted in an assembly with higher fragmentation (Figure 6B) and lower accuracy (Figure 6C, D). This data set had a read N50 of 29,404 bp, suggesting that both read length and correction contribute to assembly quality. The assembly graph for the Canu-corrected, Flye-assembled data set (visualized with Bandage [56]; Figure 7) shows six chromosome-scale contigs with smaller disjunct assembled sequences.

We used QUAST [50] to identify annotated features in our assembled sequences to compare the ability of each piece of software to assemble different classes of genomic elements. We used the WS277 annotations provided by WormBase Parasite [57] and identified exons, introns, pseudogenes, repeats, small RNAs, transposable elements (TEs) and untranslated regions (UTRs). The worst-performing assembly had >99.67% representation of exons, introns, small RNAs and UTRs and >98.85% representation of repetitive sequences such as pseudogenes, repeats and TEs (Table 2). The Flye-assembled sequences had the best representation of exons, introns, small RNAs and UTRs, while the Canu-assembled sequences had the best representation of repetitive pseudogenes, repeats and TEs.

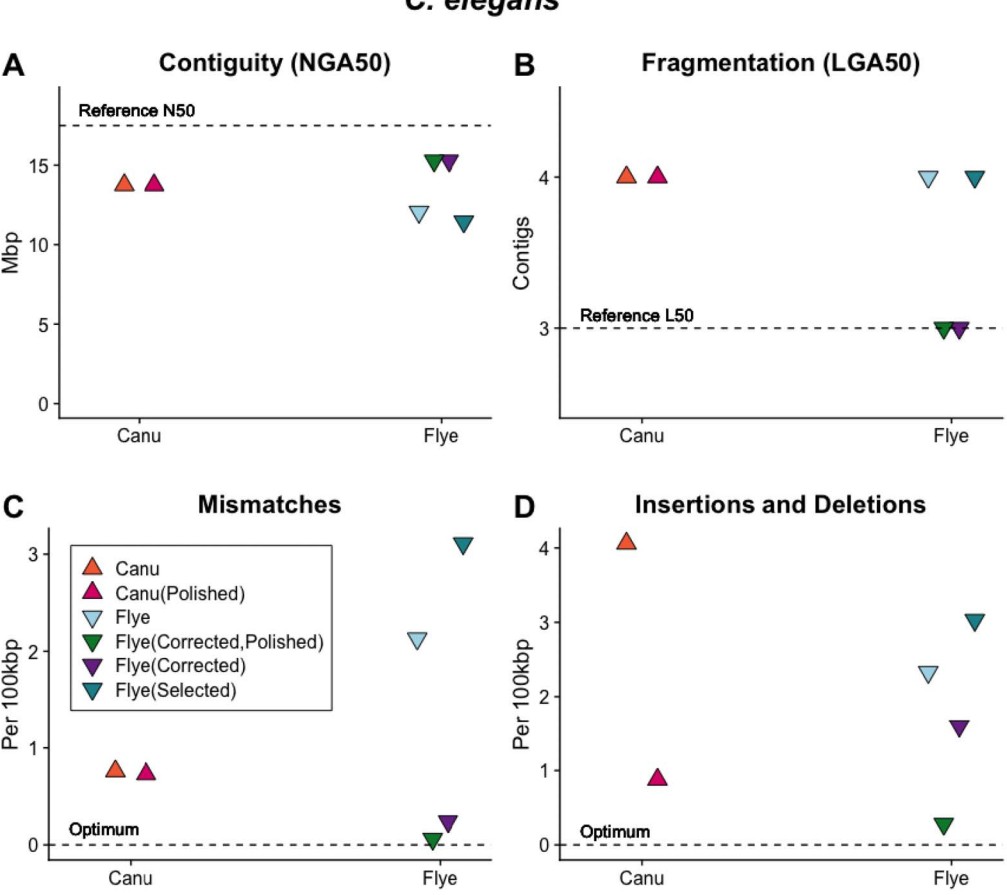

**Figure 6.** The relative performance of Canu and Flye assembly software with raw data sets, data sets with reads selected for length ("Selected"), reads selected for length and pre-corrected ("Corrected" in the figure legend) and sequences post-assembly polished with Illumina short reads ("Polished" in the figure legend) as measured by (A) genome contiguity; (B) mismatches and (C) insertions and deletions relative to the *C. elegans* reference sequence. Canu assembly of *C. elegans* ONT libraries produces assembled sequences with high contiguity, low fragmentation, few mismatches and relatively more insertions and deletions that are eliminated with post-assembly polishing. The performance of Flye was increased through read selection, correction, and post-assembly polishing with the highest-quality *C. elegans* sequence produced through Canu correction, Flye assembly and post-assembly polishing. The data set with reads selected for length did not assemble as well as that with reads selected and corrected, indicating that correction is an important step in assembly.

**Table 2.** The proportion of genomic features that QUAST identified in the assembled genome sequence.

| Assembly software | Data set | Exons | Introns | Pseudo-genes | Repeats | Small RNA | TEs | UTRs |
|---|---|---|---|---|---|---|---|---|
| Flye | Selected | **0.9988** | **0.9993** | 0.9954 | 0.9967 | **0.9985** | 0.9941 | 0.9996 |
| Flye | Corrected | **0.9988** | **0.9993** | 0.9954 | 0.9971 | 0.9984 | 0.9942 | **0.9997** |
| Flye | Corrected, Polished | **0.9988** | **0.9993** | 0.9954 | 0.9971 | 0.9984 | 0.9942 | **0.9997** |
| Canu | Corrected, Polished | 0.9986 | 0.9986 | **0.9986** | **0.9981** | 0.9981 | **0.9978** | 0.9988 |
| MaSuRCA | | 0.9967 | 0.9973 | 0.9885 | 0.9897 | 0.9972 | 0.9887 | 0.9975 |

The best performing are highlighted in bold. Flye reliably assembled exons, introns, small RNAs and untranslated regions (UTRs), while Canu performed best when assembling pseudogenes, repeats and transposable elements (TEs). Polishing did not affect the inference of genomic features.

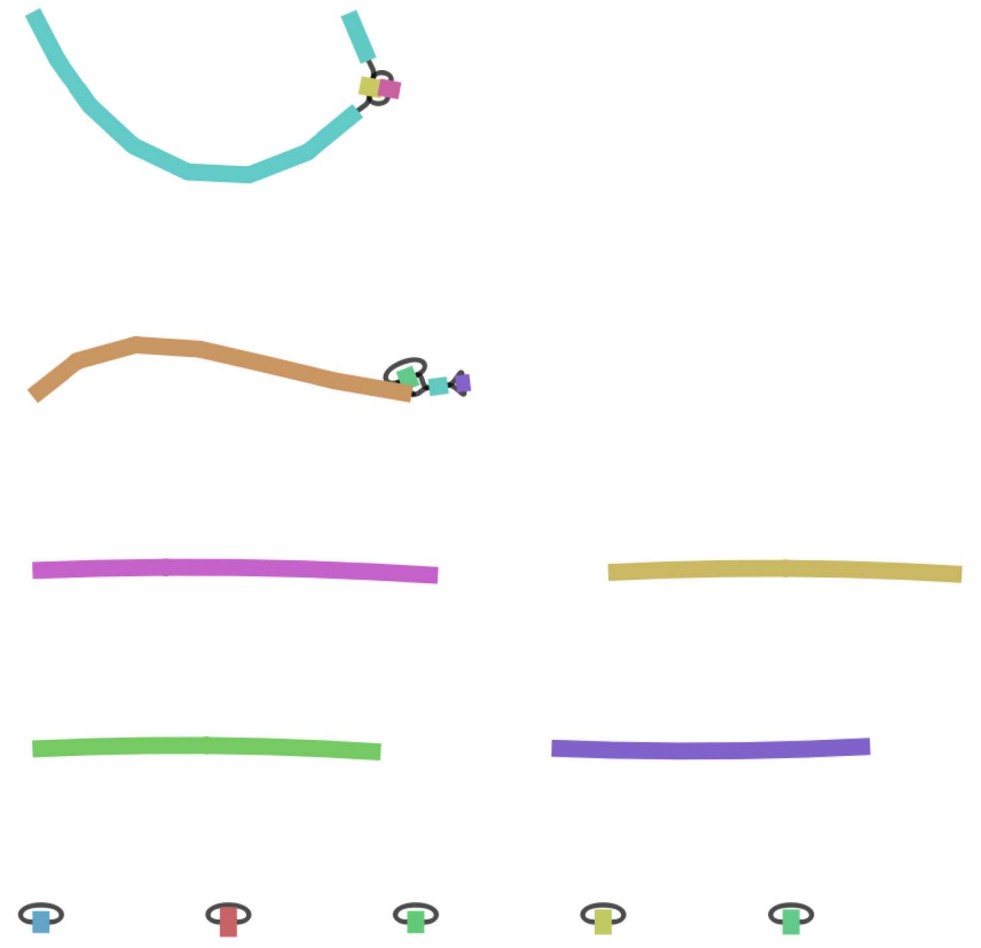

**Figure 7.** The assembly graph for the Flye assembled Canu selected and corrected *C. elegans* data set shows six large sequences corresponding to the six chromosomes of *C. elegans* with five small, unconnected sequences. Two of the large sequences contain unresolved "bubbles" in the assembly graph.

### Drosophila melanogaster

The 180-Mb genome of the diploid fruit fly *D. melanogaster* presents multiple challenges for sequencing and assembly, including approximately 60 Mb of repetitive heterochromatin [58]. The genome sequence is contained in a large X chromosome, a small ("dot") Ychromosome, and three autosomes. Assemblies of the simulated ONT *D. melanogaster* libraries repeated patterns seen with the *C. elegans* data set; the most contiguous Canu assembly was produced with ~113× ONT coverage and a read N50 of 11,679 (~16 Gb of data; Figure 8A, B). This assembly produced 145 contiguous pieces, but many of these were small. The LGA50 was four contigs (Figure 8B). The Flye assembly of the same data set was much less contiguous, producing 482 contigs. When the data set was corrected with Canu then assembled with Flye, the assembled sequences had high accuracy (Figure 9).

The top-performing Canu assembly contained 91% of the metazoan genes expected to be conserved in single copy with BUSCO [34] prior to polishing, and 94% after four rounds of Pilon with paired-end reads. In comparison, the *D. melanogaster* reference sequence

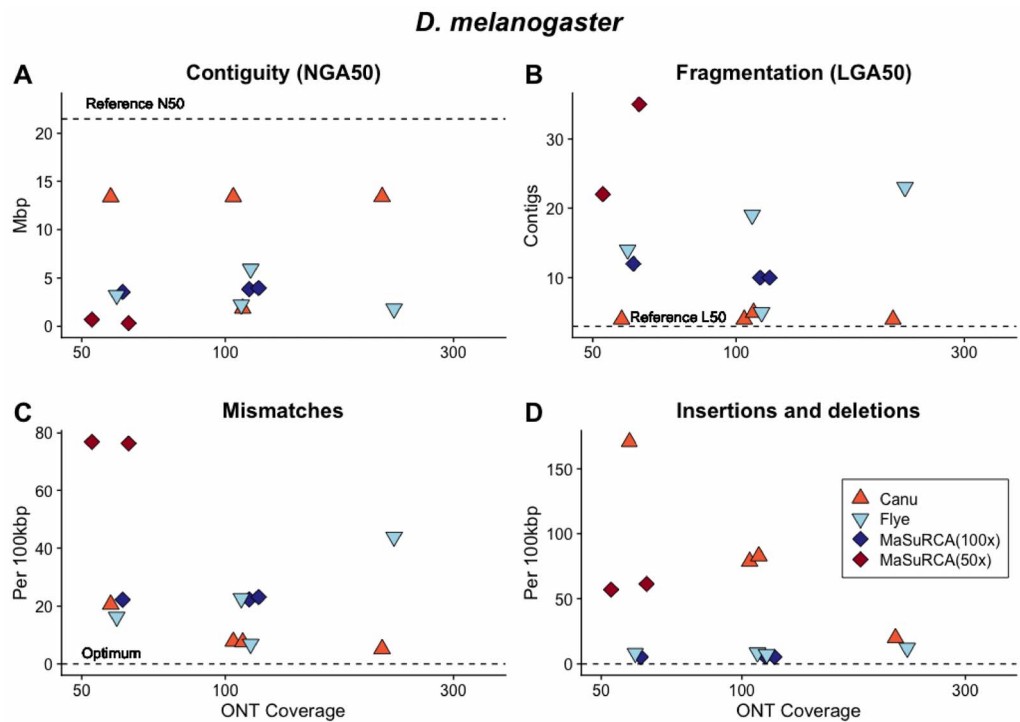

**Figure 8.** The *D. melanogaster* genome presents multiple challenges for genome sequencing and assembly, including approximately 60 Mb of repetitive heterochromatin. Canu assembly of ONT libraries with >60× sequencing depth produced a sequence with (A) high contiguity; (B) low contig number; (C) few mismatches and (D) few insertions and deletions (indels) compared with the reference. The accuracy of the Canu-assembled sequence increased with ONT library coverage. Flye assembly of the same ONT libraries produced sequences with relatively less contiguity, more fragmentation, more mismatches and fewer insertions and deletions. MaSuRCA assembly of the same ONT sequences resulted in fragmented sequences with similar error levels to the reference sequence.

contains 94.1% of the expected single copy metazoan genes. Additional data decreased contiguity, with a slight increase in accuracy and NGA50 (Figure 8A, D). Hybrid MaSuRCA [48] assemblies for *D. melanogaster* performed markedly worse than Canu assemblies of ONT data (Figure 8B). The top-performing MaSuRCA assembly produced 220 contigs, with 93.9% of the expected metazoan genes identified.

### *Arabidopsis thaliana*

The 135-Mb genome sequence of the self-fertile plant *A. thaliana* is contained in five autosomes [59]. *A. thaliana* has undergone at least two rounds of whole genome duplication [60] and contains large tracts of highly similar genomic regions. Assembly of the ONT libraries with Canu [18] and Flye [41] resulted in sequences with lower contiguity and higher fragmentation than the other model organisms, even when the library contained ~420× coverage and a read N50 of 19,577 bp (56.7 Gb of sequenced nucleotides). Canu produced 30 contiguous pieces (LGA50 five chromosomes; Figure 10) with 98.8% of the expected Viridiplantae genes [35] identified prior to polishing [34]. Flye produced 26 contigs and 98.8% of the expected Viridiplantae genes. The Flye assembly contained many more mismatches and indels than its Canu counterpart, but this discrepancy was alleviated with Pilon polishing [35]. Following polishing with Pilon [49], the Flye-assembled sequences

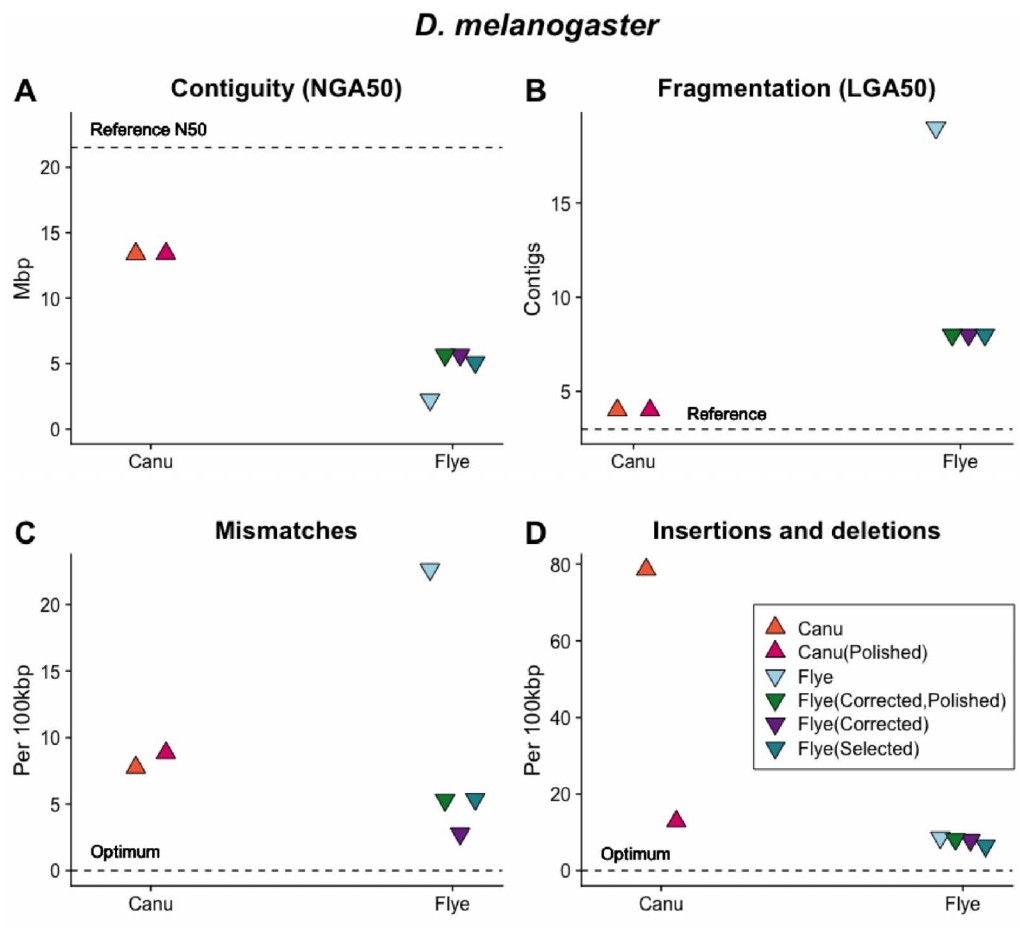

**Figure 9.** The relative performance of Canu and Flye assembly software with raw data sets, data sets with reads selected for length ("Selected"), reads selected for length and pre-corrected ("Corrected" in the figure legend) and sequences post-assembly polished with Illumina short reads ("Polished" in the figure legend) as measured by (A) genome contiguity; (B) mismatches and (C) insertions and deletions relative to the *D. melanogaster* reference sequence. Canu assembly of *D. melanogaster* ONT libraries produces assembled sequences with high contiguity, low fragmentation, few mismatches and relatively more insertions and deletions that are eliminated with post-assembly polishing. The performance of Flye can be increased by selecting, correcting and polishing.

contained 99.1% of the expected Viridiplantae genes [34], matching that of the TAIR10 reference genome for *A. thaliana*. Interestingly, our combined approach of long-read methods did not work with *A. thaliana* data sets, and neither Flye nor Canu was able to assemble a draft genome with <40× coverage.

The hybrid MaSuRCA assemblies for *A. thaliana* performed the best out of the three eukaryotes. The top-performing MaSuRCA [48] assembly produced 45 contiguous pieces, with 98.8% of the expected Viridiplantae genes identified in the sequence [34]. This was also brought up to 99.1% after polishing with Pilon [35].

### *Caenorhabditis remanei* and *C. latens*

We used our simulated data to create a protocol for *de novo* sequencing and assembly for three strains of the nematode *C. remanei* [61] and one of the closely related nematode *C. latens* [62]. Both *C. remanei* and *C. latens* are obligate outcrossing species with high levels of nucleotide diversity [63] that have hobbled previous assembly attempts [54, 64].

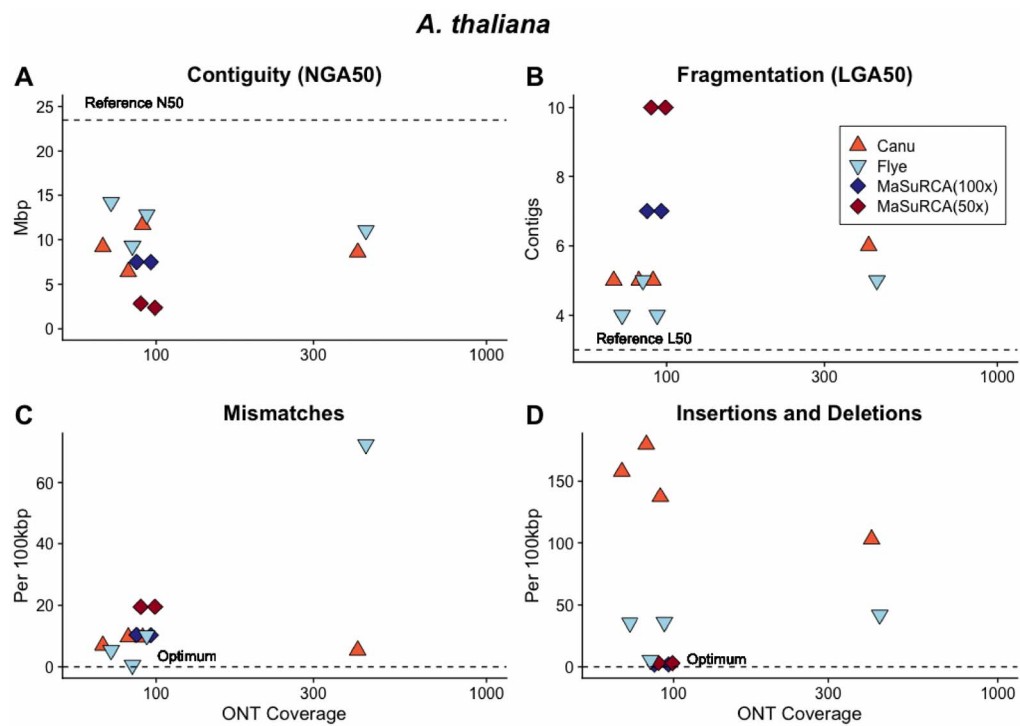

**Figure 10.** Canu, Flye and MasuRCA assembled sequences across a range of ONT sequencing coverages had (A) low contiguity; (B) high fragmentation; (C) mismatches and (D) insertions and deletions compared with the *A. thaliana* reference sequence.

The worm's small size means that pools of individuals harboring both individual and population-level diversity must be sacrificed for DNA sequencing and assembly. We prepared high-molecular-weight DNA extracts and generated ONT libraries for an outbred *C. remanei* strain EM464, two inbred laboratory strains PX356 and PX439, and the inbred laboratory *C. latens* strain PX534. We tested library preparation and computational protocols with the inbred *C. remanei* strain PX356 (Figure 11A–D) and used these to generate assembled sequences for the other *C. remanei* and *C. latens* strains.

The most contiguous *C. remanei* PX356 assembly was achieved by using Canu [18] to correct the entire ~156× coverage ONT data set prior to assembly with Flye [41]. After correction, the ~40× coverage data set had a read N50 of 21,340 bp and resulted in an assembled sequence contained in 163 contigs with 89.1% of the expected conserved nematode genes [34]. Decontamination with Blobtools2 [22] and SIDR [23] resulted in a final assembly for PX356, with 124,499,812 bp contained in 70 contigs with an N50 of 6.2 Mb and an N90 of 1.93 Mb contained in just 18 large contigs – approximately three per chromosome. We compared our results (Figures 11, 12) with a recently published sequence for the related *C. remanei* strain PX506 [40] generated with PacBio sequencing and chromatin conformation capture to produce chromosome-scale sequences [64].

Canu-only and Flye-only assemblies contained more contigs and had a smaller N50 value than the combined approach described above (Figure 12A–D). The assembly graph (Figure 13) shows several large contiguous fragments, but also multiple regions difficult to disentangle and smaller disjunct fragments. Analyses with GenomeScope [33] suggested



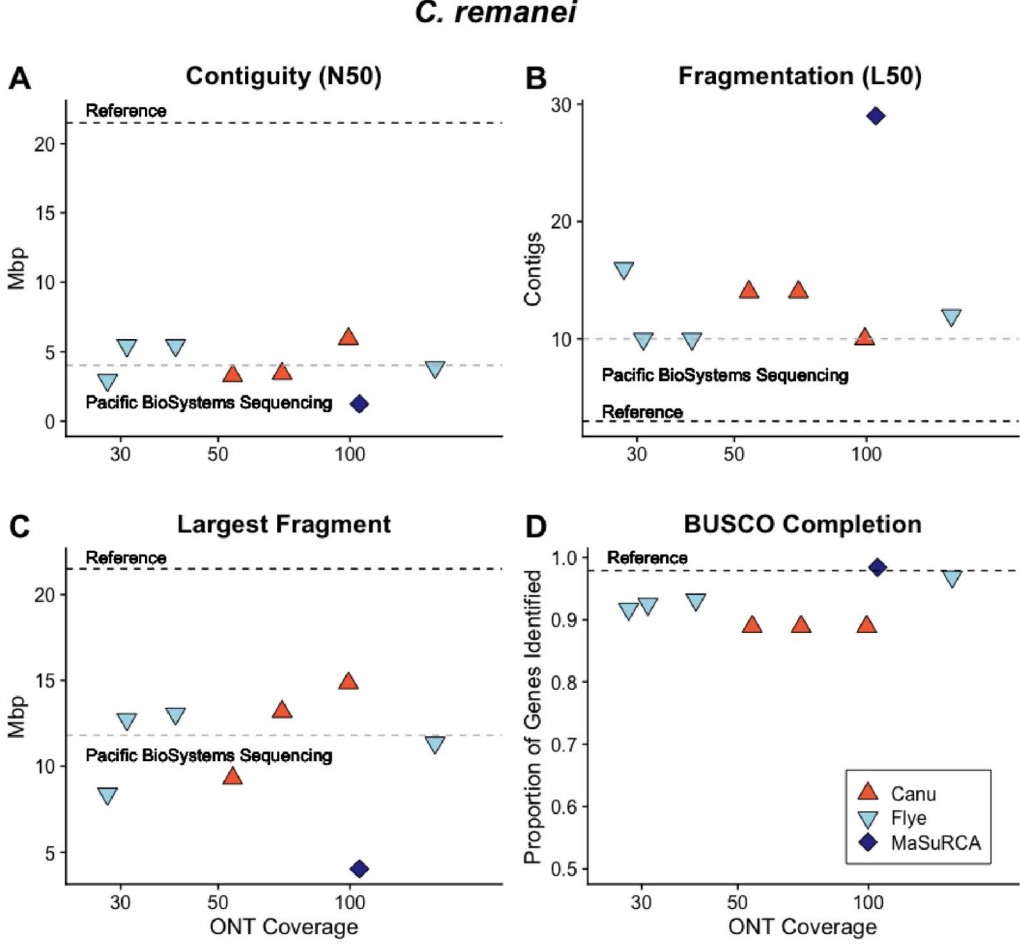

**Figure 11.** Canu, Flye and MaSuRCA-assembled *C. remanei* sequences with (A) low contiguity and (B) high fragmentation compared with the true chromosome size and number but the sequences were of comparable contiguity and fragmentation compared with Pacific BioSystems sequences. The (C) largest fragment was also comparable to that achieved by Pacific BioSystems and the (D) BUSCO genome completeness was high with assembly of the highest coverage ONT library.

~0.7% of the genome remains heterozygous (Figure 14). Assemblies with subsamples of the long-read data were increasingly fragmented with decreasing coverage. Flye assemblies with initial coverages of 102×, 70×, and 52× produced 183, 242, and 267 contigs, respectively. A MaSuRCA-hybrid approach [48] using 102× ONT coverage and 450× paired-end coverage yielded 336 contigs and 96.6% of BUSCO single-copy genes.

BUSCO identified 97.7% (increased from 89.1%) of the expected conserved single copy genes following four rounds of polishing using Pilon [49] with Illumina paired-end reads at ~225× average depth (Figure 8). We tested the influence of Illumina coverage on polishing and found that 97.6% of the expected conserved nematode genes could be identified after polishing with just ~20× Illumina coverage (Table 3), indicating that a large amount of data is not necessary to correct most errors in an assembly. However, this was achieved after three successive rounds of error correction with the Pilon software package [49] utilizing the same Illumina DNA sequence reads.

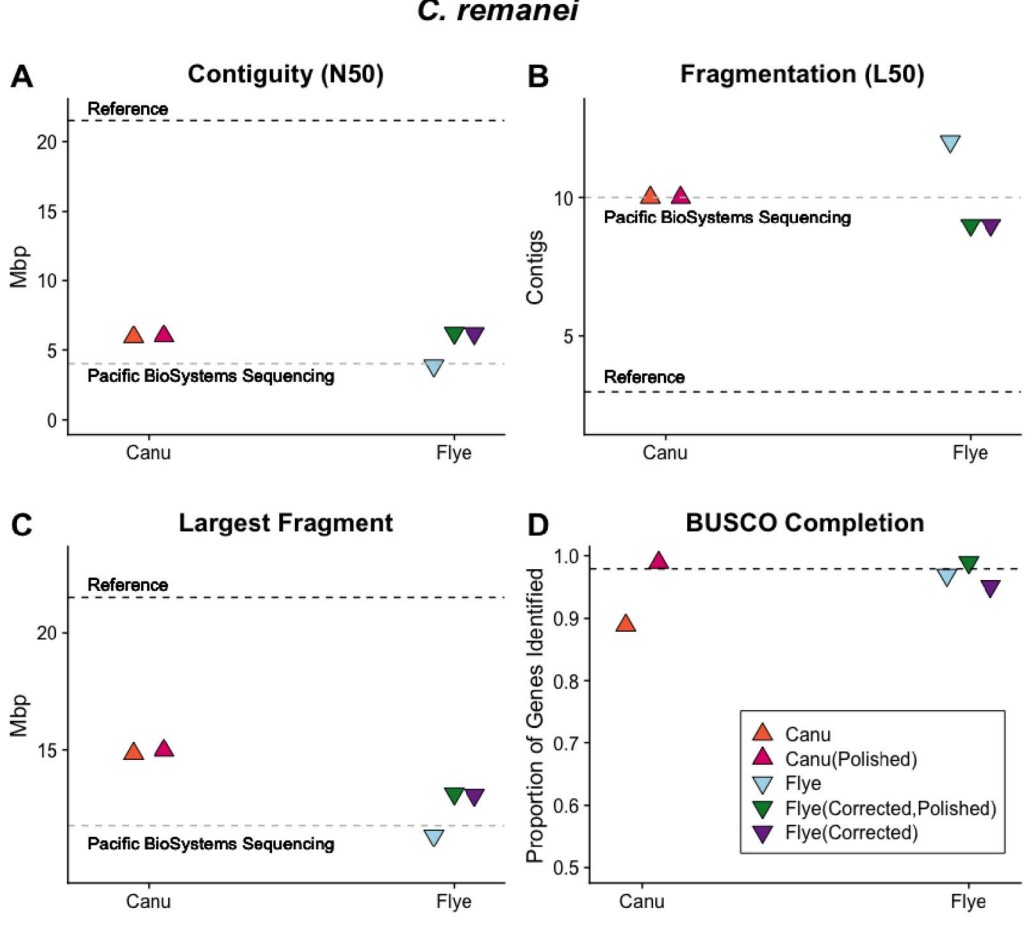

**Figure 12.** Both Canu and Flye produced assembled sequences with (A) contiguity; (B) fragmentation; (C) the largest fragment and (D) BUSCO gene completeness comparable to Pacific BioSystems sequencing after read correction and post-assembly polishing.

We generated assembled genome sequences for *C. remanei* PX439, *C. remanei* EM464, and *C. latens* PX534 using the "best practices" protocols developed through simulations and our *C. remanei* PX356 experimentation. Our *C. remanei* PX439 data set had an ONT coverage depth of ~120× with a read N50 of 18,233 bp. After Canu correction, the read N50 increased to 38,434 bp and we assembled a 132,046,643-bp genome sequence contained in 53 contigs (N50 12.6 Mb). The N90 for this assembled sequence was 2.27 Mb contained in 12 large contigs, approximately two per chromosome.

*C. remanei* EM464 is outbred and retains high levels of heterozygosity. We did not expect to assemble a cohesive genome sequence, but generated ONT libraries and an assembled sequence to identify the distribution of allelic variants in the population. The ONT library had a coverage depth of ~76× and a read N50 of 22,891 bp. The Canu-corrected data set used for Flye assembly had an N50 of 34,951 bp. Flye produced a 138,626,018-bp genome sequence contained in 320 contigs (N50 3.4 Mb). The flow cytometry-estimated genome size of *C. remanei* is 124–131 Mb, and we estimate that 5–12% of the assembled sequences are allelic variants. The *C. latens* PX534 data set had a coverage depth of ~119× and a read N50

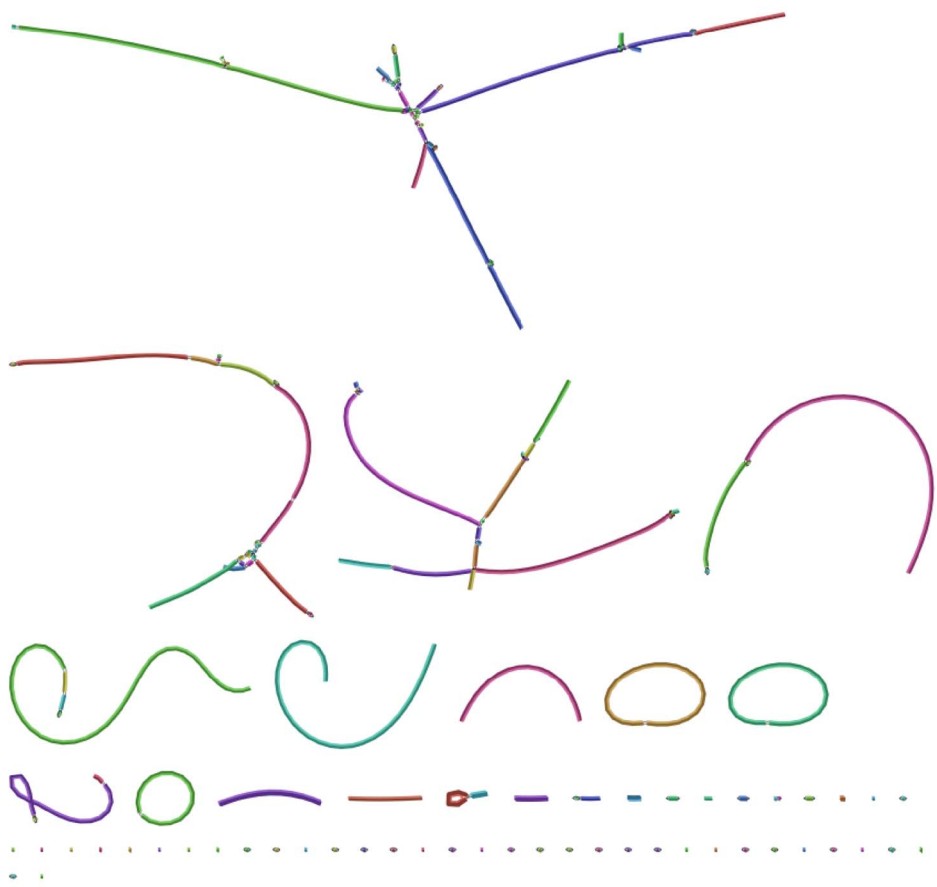

**Figure 13.** The *C. remanei* PX356 assembly graph shows multiple unresolved regions, bacterial contaminants (visible as circular chromosomes) and small fragments.

of 14,759 bp. After Canu correction, the read N50 of the data set was 34,591 bp, and we assembled a 120,367,464-bp genome sequence in 97 contigs with an N50 of 13.6 Mb.

## DISCUSSION

We have found that Canu [18] and Flye software packages [41] produce high-quality contiguous draft assemblies from ONT libraries for eukaryotes with moderate genome sizes. In many cases, Flye slightly outperforms Canu, while being much faster and using fewer computational resources [12, 44]. We found that, given the same set of corrected reads, Flye regularly produces a more contiguous, higher accuracy assembled sequence when compared with Canu. This finding highlights the need for performing multiple assembly strategies to identify the top performing strategy for each data set.

The hybrid assemblies produced by MaSuRCA [48] contained a higher proportion of expected conserved genes than the unpolished long-read only assemblies, and had fewer mismatches and insertion/deletion errors. However, given the same amount of ONT data, both Canu [18] and Flye [41] assembled more contiguous genome sequences. The long-read-only assemblies also produced larger NG50 values and contained a larger fraction of the expected genome than hybrid MaSuRCA assemblies. The shortfalls of long-read-only assembly can be overcome by "polishing" with ~20× Illumina DNA

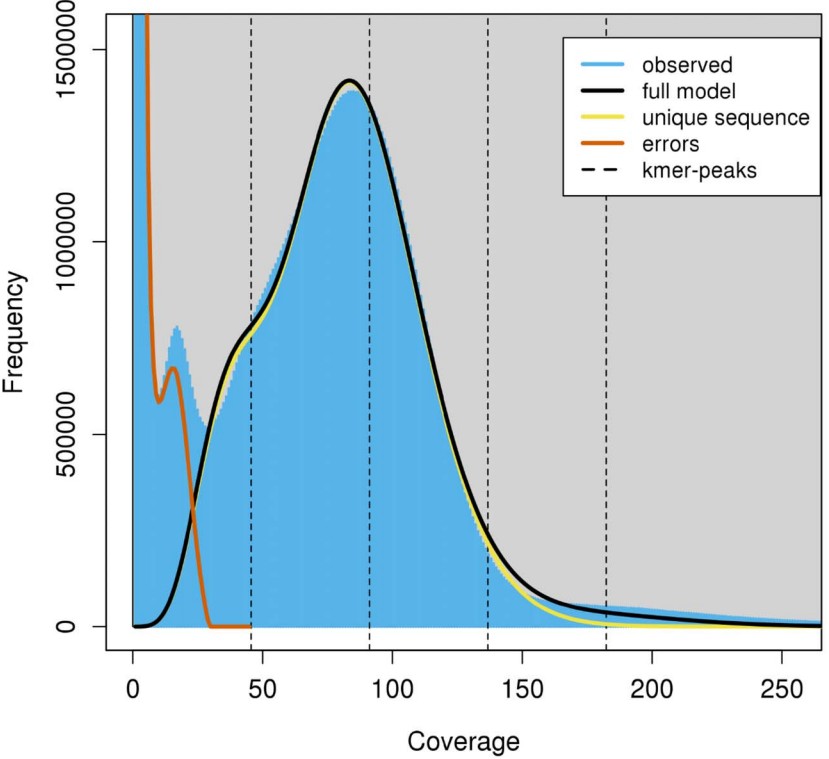

## GenomeScope Profile
len:108,521,241bp uniq:84.6% het:0.712% kcov:45.6 err:0.749% dup:5.16% k:21

**Figure 14.** Analysis with the GenomeScope software revealed residual heterozygosity in the *C. remanei* PX356 sequences (estimated at 0.712%). Here, the residual heterozygosity is visible as a shoulder at ~50× sequencing coverage.

**Table 3.** Polishing with low, medium and high coverage Illumina data.

| | Polishing | | | BUSCO | | |
|---|---|---|---|---|---|---|
| **Sample** | **Rounds** | **Coverage (×)** | **Single (%)** | **Duplicated** | **Fragmented** | **Missing** |
| Canu | 0 | 20 | 80.3 | 0.9 | 7.7 | 11.1 |
| Assembled | 1 | 20 | 97.2 | 1 | 0.5 | 1.3 |
| *C. remanei* | 2 | 20 | 97.3 | 1 | 0.4 | 1.3 |
| | 3 | 20 | 97.6 | 1 | 0.2 | 1.2 |
| | 4 | 20 | 97.6 | 1 | 0.2 | 1.2 |
| | 1 | 114 | 97.6 | 1.1 | 0.3 | 1 |
| | 2 | 114 | 97.6 | 1 | 0.3 | 1.1 |
| | 3 | 114× | 97.5 | 1 | 0.3 | 1.2 |
| | 4 | 114 | 97.5 | 1 | 0.3 | 1.2 |
| | 1 | 227 | 97.6 | 1 | 0.3 | 1.1 |
| | 2 | 227 | 97.5 | 1 | 0.3 | 1.1 |
| | 3 | 227 | 97.7 | 0.9 | 0.3 | 1.1 |
| | 4 | 227 | 97.7 | 0.9 | 0.3 | 1.1 |

Polishing the *C. remanei* assembled sequence with an Illumina library at 20× depth increases the percentage of conserved genes identified by BUSCO [34] after three rounds of polishing with Pilon [49]. Polishing with a higher depth Illumina library (114× average sequencing coverage) produces similar results after two rounds of polishing with Pilon. The highest percentage of conserved genes found in single copy is achieved after three rounds of polishing with Pilon and a high depth Illumina library at 227× average coverage.

**Table 4.** Library statistics and BUSCO completeness for the *Caenorhabditis* assembled sequences.

| Organism | Assembly version | ONT coverage | Read N50 | Post-correction N50 | Assembly size (Mb) | Contigs | Single copy | Duplicate | Fragmented | Missing |
|---|---|---|---|---|---|---|---|---|---|---|
| | | | | | | | | **BUSCO** | | |
| *C. remanei* PX356 | Published PX356 | — | — | — | 145.44 | 1,591 | | | | |
| | Draft PX356 | 156 | 8,203 | 21,340 | 154.64 | 163 | | | | |
| | Final PX356 | | | | 124.5 | 70 | 97.7 | 1 | 0.2 | 1.1 |
| *C. remanei* PX439 | Published PX439 | — | — | — | 124.54 | 912 | | | | |
| | Draft PX439 | 95 | 18,233 | 38,434 | 144.15 | 85 | | | | |
| | Final PX439 | | | | 131.75 | 53 | 97.6 | 0.9 | 0.3 | 1.2 |
| *C. remanei* EM464 | Published EM464 | — | — | — | N/A | N/A | | | | |
| | Draft EM464 | 76 | 22,891 | 33,688 | 154.81 | 400 | | | | |
| | Final EM464 | | | | 138.62 | 320 | 96.2 | 2.6 | 0.3 | 0.9 |
| *C. latens* PX534 | Published PX534 | — | — | — | 117.12 | 1,858 | | | | |
| | Draft PX534 | 119 | 14,759 | 34,951 | 125.13 | 132 | | | | |
| | Final PX534 | | | | 124.94 | 98 | 97.6 | 1.1 | 0.3 | 1 |

Accessions: PX356, GCA_001643735.2; PX439, GCA_002259225.1; EM464, N/A; PX534, GCA_002259235.

sequences using software such as Pilon [49] to error-correct the draft assemblies. We found that polishing was more important for increasing accuracy of the real-world data than the results seen with simulated data sets. These methods improved the accuracy of Canu or Flye assemblies to be on par with, or better than, those produced by MaSuRCA.

These highly contiguous assemblies were achieved with relatively high sequencing depth; often >100× coverage across the genome. Although ONT can theoretically produce megabase-sized reads, in reality, many of the sequence reads in "real" projects are shorter because of fragmentation that occurs during DNA purification and library preparation. ONT libraries may have many more small reads and >100× "real" coverage may be necessary to achieve >40× coverage with reads >25 Kb and highly contiguous assemblies. Increasing read N50 in a library through improvement of DNA isolation and library preparation methods reduces the necessary coverage to produce a high-quality assembly [66]. Therefore, researchers should adjust their coverage goals based on the quality of libraries they can reliably produce.

Our findings with simulated data sets were supported by all real-world data sets tested. *Caenorhabditis remanei* PX356, PX439 and *C. latens* PX534 have assembled sequences generated with paired-end Illumina sequences [55]. Our assembled sequences were at least one order of magnitude more contiguous than previous attempts, highlighting the ability of ONT long reads to improve contiguity. Outbreeding *Caenorhabditis* are resistant to inbreeding [67] and previous assembly attempts resulted in over-assembly due to residual heterozygosity [64] or under-assembly. Despite this, we finalized a highly contiguous and complete assembly for these nematodes (Table 4). The strains we targeted are closely related and interfertile [68], yet their assembled genome sequences vary by as much as 10% in length. This level of structural variation between closely related species highlights the need for *de novo* assembled sequences in molecular evolution research.

The findings we present here clearly demonstrate that existing genome sequences can be improved by adding in ONT data sets. ONT is continually increasing accuracy through library preparation, and a new "Q20" sequencing chemistry kit is currently in the hands of developers. A high accuracy "bonito" basecaller is in development [69] and new software

like NECAT [70] and NextDenovo/NextPolish [71] are being developed to specifically address ONT error structures. As sequencing and assembly technologies continue to improve and reduce in cost, many researchers will find published data sets can be updated to further improve downstream analyses.

## REUSE POTENTIAL

Considering our findings, we suggest that long-read data is prioritized when undertaking *de novo* genome assembly projects. Our results indicate that an assembly with sufficient ONT long read coverage will be highly contiguous, and polishing with Illumina data can achieve high levels of accuracy. For situations in which high-coverage ONT libraries are not feasible, MaSuRCA-assembled [48] Illumina and ONT read sets can produce reliable draft sequences. However, the quality and contiguity of the assembled sequence is determined by Illumina read depth and effort should be made to increase Illumina read depth, even if it is at the expense of ONT sequences. MaSuRCA-assembled Illumina sequences have fewer mismatches and insertion/deletion errors than MaSuRCA-assembled ONT and Illumina hybrid read sets, indicating that the inclusion of ONT sequences introduces errors. We suggest that error correction with Illumina DNA sequences and the Pilon software package [49] is a necessary finishing step in any assembly project utilizing ONT data.

Our results demonstrate that near-chromosome-level genome sequences are achievable with sufficient ONT data. However, chromosome-level genome assemblies are often not necessary to address many research questions, particularly those focused on small numbers of genes, phylogenomic information or population variation. Researchers should approach genome sequencing by first determining what genome-completion level will be sufficient for their research goals. To aid in this approach, we hope our study will help researchers determine the amount of sequencing effort, and the sequencing approaches, that will best suit their needs.

## DATA AVAILABILITY

The data sets supporting the results of this article are available in the *GigaScience* repository, GigaDB [35] and Dryad Digital Repository [72]. Accessions are available under the following accession numbers: *E. coli* genome sequence GCF_000005845.2, ONT read set SRR8154670; *C. elegans* genome sequence GCA_000002985.3, ONT read set ERR2092776; *A. thaliana* genome sequence GCA_000001735.1, ONT read set ERR2173373; *D. melanogaster* genome sequence GCA_000001215.4, ONT read set SRR6702603; *C. remanei* PX356 BioProject PRJNA248909; *C. remanei* PX439 BioProject PRJNA248911; *C. remanei* EM464 PRJNA562722; *C. latens* PX534 BioProject PRJNA248912.

## DECLARATIONS
## LIST OF ABBREVIATIONS

bp: base pair; BUSCO: Benchmarking Universal Single-Copy Ortholog; gDNA: Gb: gigabase pair; GB: gigabyte; genomic DNA; Kb: kilobase pair; MaSuRCA: Maryland Super Read Cabog Assembler; Mb: megabase pair; NCBI: National Center for Biotechnology Information; ONT: Oxford Nanopore Technologies; PacBio: Pacific Biosciences; SIDR: Sequence-based Identification with Decision tRees; TE: transposable element; UTR: untranslated region.

## ETHICAL APPROVAL

Not applicable.

## CONSENT FOR PUBLICATION
Not applicable.

## COMPETING INTERESTS
The authors declare that they have no competing interests.

## FUNDING
This work is funded by NSF uRoL 1921585 and NSF DEB 1941854 to Janna L. Fierst.

## AUTHORS' CONTRIBUTIONS
JMS and JLF conceived the study and developed the methodology; JMS, JDM, ACM and JLF conducted formal analysis; JLF developed resources; JMS, JDM and JLF wrote the original draft, and reviewed and edited the manuscript; JLF was responsible for visualization and supervision.

## ACKNOWLEDGEMENTS
We thank Paula Adams, Denise Akob, Louis Bubrig, Rebecca Varney and Kevin Kocot for helpful discussions. Patrick C. Phillips generously provided the PX356, PX439 and PX534 inbred strains and the Caenorhabditis Genetics Center provided the EM464 outbred strain.

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
