## [Reviewer Report]

Reviewer name and names of any other individual's who aided in reviewerZhao ChenDo you understand and agree to our policy of having open and named reviews, and having your review included with the published manuscript. (If no, please inform the editor that you cannot review this manuscript.)YesIs the language of sufficient quality?NoPlease add additional comments on language quality to clarify if neededThere are many grammatical errors and typos that should be corrected.Is there a clear statement of need explaining what problems the software is designed to solve and who the target audience is? YesAdditional CommentsIs the source code available, and has an appropriate Open Source Initiative license <a href="https://opensource.org/licenses" target="_blank">(https://opensource.org/licenses)</a> been assigned to the code?YesAdditional CommentsAs Open Source Software are there guidelines on how to contribute, report issues or seek support on the code?YesAdditional CommentsIs the code executable?YesAdditional CommentsIs installation/deployment sufficiently outlined in the paper and documentation, and does it proceed as outlined?YesAdditional CommentsIs the documentation provided clear and user friendly?YesAdditional CommentsIs there a clearly-stated list of dependencies, and is the core functionality of the software documented to a satisfactory level?YesAdditional CommentsHave any claims of performance been sufficiently tested and compared to other commonly-used packages? YesAdditional CommentsAre there (ideally real world) examples demonstrating use of the software? YesAdditional CommentsAdditional CommentsAny Additional Overall Comments to the AuthorThe authors should clarify why only Canu and Flye were selected instead of other long-read assemblers such as Raven, Redbean, Shasta, and Miniasm. Rationales should be given for why these two assemblers were selected. The same thing for MaSuRCA. It looks like you used MaSuRCA for hybrid assembly. Unicycler also contains a commonly used hybrid assembly pipeline. Therefore, you should also explain why MaSuRCA was selected for your study.
A flow chart with all bioinformatics tools included should be provided to show more clearly how this entire study was carried out, including assembly, error correction, and analysis after assembly.
More information about the quality of long reads should be provided, such as Phred quality scores, percentage of reads with Q30 or above, and average read lengths. QUAST should suffice for these quality analyses.
Only testing simulated reads is not sufficient for making a solid conclusion since simulated reads cannot be treated as being equal to real reads or reflect basecalling errors in real reads. Since real reads are readily available on NCBI, real reads should also be tested. As your title didn’t mention anything about the fact that this study was solely based on testing simulated reads and your objective was to optimize the bioinformatic pipeline for processing Oxford Nanopore long reads, the experiments should be performed by including all conditions. Accordingly, real reads should also be tested, which could significantly improve the scientific quality of this study.
Line12-13: This may not be true, since many studies have been published on how to assemble and error-correct Oxford Nanopore long reads to produce accurate genomes. The authors should describe why the present study is novel and what new findings were reported.RecommendationMajor Revisions

---

## [Reviewer Report]

Reviewer name and names of any other individual's who aided in reviewerShanlin LiuDo you understand and agree to our policy of having open and named reviews, and having your review included with the published manuscript. (If no, please inform the editor that you cannot review this manuscript.)YesIs the language of sufficient quality?YesPlease add additional comments on language quality to clarify if neededIs there a clear statement of need explaining what problems the software is designed to solve and who the target audience is? YesAdditional CommentsIs the source code available, and has an appropriate Open Source Initiative license <a href="https://opensource.org/licenses" target="_blank">(https://opensource.org/licenses)</a> been assigned to the code?YesAdditional CommentsAs Open Source Software are there guidelines on how to contribute, report issues or seek support on the code?YesAdditional CommentsIs the code executable?YesAdditional CommentsIs installation/deployment sufficiently outlined in the paper and documentation, and does it proceed as outlined?YesAdditional CommentsIs the documentation provided clear and user friendly?YesAdditional CommentsIs there a clearly-stated list of dependencies, and is the core functionality of the software documented to a satisfactory level?YesAdditional CommentsHave any claims of performance been sufficiently tested and compared to other commonly-used packages? YesAdditional CommentsAre there (ideally real world) examples demonstrating use of the software? YesAdditional CommentsIs automated testing used or are there manual steps described so that the functionality of the software can be verified?YesAdditional CommentsAny Additional Overall Comments to the AuthorThe genome de novo assembly based on third generation sequencing (ONT in the current work) has been widely applied for plenty of organisms, including bacteria, plants and animals with various genome sizes, e g. the two recently published lungfish genomes (genome size of > 30 G) in Nature and Cell, and genomes of a broad range of species published in GigaScience, Scientific Data, Molecular Ecology Resources, et al. It is pretty easy to find the analysis pipeline or datasets that were used to obtain a high quality genome assembly in those published works. The authors generated multiple genome assemblies for four model species using different simulated datasets with varied sequencing depths and different assembly tools, and tried to provide useful guidance for those who are new to genome assembly. However, I am afraid that the current study contains some limitations in the results and conclusions that may mislead the readers, and I do recommend the authors reconstruct the manuscript and address those issues before its publication.
First of all, a routine practice of genome assembly with long reads (either ONT or PacBio) includes a polishing step based on long reads itself using tools like Nanopolish, Medaka, Racon, et al. The author skipped this step in all of their analyses and directly evaluated the assembly errors based on the outputs generated from different combinations of datasets and software. It has little practical value whatever the results showed.
Secondly, the four model species included in the current work can hardly represent a broad range of organisms – all have a genome size < 200 MB and low level of repetitive elements (< 30%). Hence, the analysis results from the current work offer scant guidance to those who work on organisms like plants, fishes, insects, mammals et al. For example, computing resources become the first hurdle for the genome assembly when working on > 100X ONT reads for the species with large genome size even if you can afford the sequencing. So, researchers would generate less data or prefer assemblers like WTDBG, NextDenovo, Falcon to obtain their genome assembly. In addition, the authors deem Caenorhabditis species as a highly heterozygous genome (0.7% according to their calculation), which is also open to question. Genomes of multitudinous insects and plants have a much higher heterozygous level.
What’s more, the authors may want to pay attention to the news regarding the Sequel II sequencing platform recently released by PacBio Tech. As far as I know, it can provide inexpensive long read sequencing thanks to its huge improvement in sequencing throughput. Also, it also has a new release of a library preparation kit that can work on low amounts of DNA inputs. If so, what you stated in the instruction section may be incorrect.
Beside the major issues mentioned above, there are some other minor ones listing as follows:
Line 89. The authors may want to provide common names of those model species to improve readability of the manuscript.
Line 119 Genome references and ONT reads were derived from different individuals or strains, and there are very low coverage ONT reads for E. coli. I am not sure whether those factors will influence the quality of simulation or not. The authors may add a caution to clarify these concerns.

Line 24 A combination of experimental techniques? It is better to specify what experimental techniques.
Line 128 Incorrect word format and C. latens missed.
Line 141 How to define the best performance, the most contiguous assembly?
Line 137 When you say “failed to produce an assembly”, does it mean that software failed to generate outputs or unexpected assembly results?
Line 287 Supplement the BUSCO value of the reference TAIR10
Line 287 what do you mean by “combined approach”? Do you mean the method that corrects reads using Canu and assembles them using FLYE?
Line 233 – 241 the “corrected” and “selected” dataset used in the Nematoda test were not applied to other organisms.
Line 241 Canu correction could truncate some low quality reads or cut long reads into multiple pieces for speculated chimeric reads. I don’t think you can reach a conclusion that read length influences assembly quality using the current results.
Line 242 Please rephrase this sentence and put Figure 5 and reference #36 in better positions to avoid misunderstanding.
Line 337 – 341 duplicates to the content line 308 – 312, and conflicts between each other.
Line 355 All the tested organisms have genome sizes < 200 MB, please specific this limitation instead of saying a broad range of organisms.
Line 368 Low coverage may mislead readers, the authors cannot reach such a conclusion based on merely one single test.
Line 461 which model was used – high accuracy? or flip-flop?
Table 1. Too long a header, could move some of the content as table notes.
RecommendationMajor Revisions